# The Influence of Capping Layers on Tunneling Magnetoresistance and Microstructure in CoFeB/MgO/CoFeB Magnetic Tunnel Junctions upon Annealing

**DOI:** 10.3390/nano13182591

**Published:** 2023-09-19

**Authors:** Geunwoo Kim, Soogil Lee, Sanghwa Lee, Byonggwon Song, Byung-Kyu Lee, Duhyun Lee, Jin Seo Lee, Min Hyeok Lee, Young Keun Kim, Byong-Guk Park

**Affiliations:** 1Department of Materials Science and Engineering, Korea Advanced Institute of Science and Technology, Daejeon 34141, Republic of Korea; 2Department of Electronic Engineering, Gachon University, Seongnam 13120, Republic of Korea; 3Samsung Advanced Institute of Technology, Samsung Electronics, Suwon 16678, Republic of Korea; 4Department of Semiconductor Systems Engineering, Korea University, Seoul 02481, Republic of Korea; 5Department of Materials Science and Engineering, Korea University, Seoul 02481, Republic of Korea

**Keywords:** capping layer, magnetic tunnel junction, tunneling magnetoresistance, diffusion

## Abstract

This study investigates the effects of annealing on the tunnel magnetoresistance (TMR) ratio in CoFeB/MgO/CoFeB-based magnetic tunnel junctions (MTJs) with different capping layers and correlates them with microstructural changes. It is found that the capping layer plays an important role in determining the maximum TMR ratio and the corresponding annealing temperature (*T*_ann_). For a Pt capping layer, the TMR reaches ~95% at a *T*_ann_ of 350 °C, then decreases upon a further increase in *T*_ann_. A microstructural analysis reveals that the low TMR is due to severe intermixing in the Pt/CoFeB layers. On the other hand, when introducing a Ta capping layer with suppressed diffusion into the CoFeB layer, the TMR continues to increase with *T*_ann_ up to 400 °C, reaching ~250%. Our findings indicate that the proper selection of a capping layer can increase the annealing temperature of MTJs so that it becomes compatible with the complementary metal-oxide-semiconductor backend process.

## 1. Introduction

Magnetic tunnel junctions (MTJs) consisting of ferromagnet (FM)/tunnel barrier/FM structures have been studied extensively because they serve as a key element in various spintronic devices, including the read heads of hard disk drives and magnetic random-access memory devices (MRAMs) [1,2,3,4,5]. In MTJs, tunneling electrons are spin-polarized along the magnetization direction so that the tunneling probability increases (decreases) when the magnetizations of the two FMs are aligned in a parallel (antiparallel) configuration. This spin-dependent tunneling can be enhanced by introducing a (001)-oriented crystalline MgO barrier, where the Δ_1_ Bloch state allows coherent tunneling [6,7,8]. A large tunneling magnetoresistance (TMR) ratio exceeding hundreds of percent as theoretically predicted in crystalline MgO-based MTJs has been experimentally demonstrated [9,10,11,12].

CoFeB is a widely employed FM electrode in sputter-grown MTJs [11,12,13,14,15,16,17,18,19,20] because, first, amorphous CoFeB favors the growth of the MgO close-packed (001) plane on top [12,15], and second, post-annealing causes boron to diffuse out of CoFeB and subsequently crystallize the amorphous CoFeB into CoFe on the MgO (001) texture [19,20]. It is also known that post-annealing at moderate temperatures improves the material and device properties of the MTJs by reducing defects in the MgO tunnel barrier or ferromagnet/MgO interfaces and/or by enhancing the crystallinity of the MgO layer. This leads to an increase in the TMR ratio or the perpendicular magnetic anisotropy (PMA). Therefore, in order to obtain a large TMR ratio in CoFeB/MgO/CoFeB-structured MTJs, it is essential to perform post-annealing at an elevated temperature. However, the annealing temperature (*T*_ann_) must not be excessive such that it causes atomic intermixing between the layers, which diminishes the TMR. Furthermore, for MTJs to be used in practical MRAM applications, they must maintain their characteristics at *T*_ann_ values above 400 °C, the range in which complementary metal-oxide-semiconductor (CMOS) backend integration takes place [21,22,23].

Many studies have examined annealing effects in antiferromagnet/CoFeB/MgO/CoFeB structures and the interfaces therein [9,12,16,19,24,25,26,27,28,29,30,31,32,33,34,35,36,37,38,39,40,41]. For example, the effect of the Ta capping layer has been extensively investigated. Upon annealing, the Ta capping layer effectively absorbs boron atoms from adjacent CoFeB layers, promoting the crystallization of the amorphous CoFeB layer. Consequently, MTJs with a Ta capping layer exhibit enhanced TMR ratios after annealing within a *T*_ann_ range of 300~500 °C [12,19,28,29,30,31,32,33,34,35]. The W or Hf capping layer demonstrates similar annealing effects in terms of the increased TMR because of the effective B absorption [36,37,38]. On the other hand, there have been few studies of the effects of annealing on Pt/CoFeB layers [39,40] or a Pt/CoFeB/MgO/CoFeB tunnel junction [41], where PMA deteriorates when samples are annealed with *T*_ann_ exceeding 300 °C. This was attributed to intermixing between the Pt and CoFeB layers. However, previous studies did not investigate the annealing effect on the TMR ratio in MTJs with a Pt capping layer, especially at the *T*_ann_ representative of the CMOS-compatible backend process.

Furthermore, heavy-metal (e.g., Pt, Ta, W)/ferromagnet bilayers have been extensively investigated because the spin currents generated in the heavy metal exert spin-orbit torque (SOT) on the ferromagnet and control its magnetization direction. SOT is being developed as a novel writing technology for energy-efficient MRAM [42,43,44,45]. Among such heavy elements, Pt is considered an excellent SOT material owing to its low resistivity and relatively large spin Hall angle, offering a distinct advantage in terms of power consumption for SOT-based spintronic devices over other spin-current source materials. Therefore, it is crucial to investigate the annealing effect on the TMR of CoFeB/MgO/CoFeB MTJs with a Pt capping layer compared to the effects on those with other capping layers.

In this study, we investigated the annealing-temperature dependence of TMR in IrMn/CoFeB/MgO/CoFeB MTJs with different heavy-metal capping layers of Pt, Ta, and W and correlated them with microstructural changes. For a Pt capping layer, the TMR reaches its maximum value of ~95% at *T*_ann_ of 350 °C and is then reduced as *T*_ann_ is increased further. Microstructural analyses reveal that annealing causes severe interdiffusion between the Pt and CoFeB layers, which is believed to be responsible for the reduced TMR value. Interestingly, when Ta is used as a capping layer, the TMR increases to ~250% and does not deteriorate even at a *T*_ann_ of 400 °C, meeting the thermal budget requirements of the CMOS backend process. MTJs with a W capping layer also exhibit similar behavior. Microstructural analyses confirm that intermixing is significantly suppressed in the CoFeB/Ta layer. Our study indicates that an appropriate capping layer can enhance the temperature dependence of the TMR in CoFeB/MgO/CoFeB MTJs.

## 2. Experimental Section

We fabricated MTJs consisting of a Ni_81_Fe_19_ (1 nm)/Ir_25_Mn_75_ (15 nm)/Co_32_Fe_48_B_20_ (5 nm)/MgO (2 nm)/Co_32_Fe_48_B_20_ (4 nm) structure with different capping layers of Pt, Ta, and W, as illustrated in Figure 1a. The films were deposited on Si/SiO_x_ (200 nm) substrates by magnetron sputtering at room temperature with a base pressure lower than 3.0 × 10^−8^ Torr. During the deposition, a magnetic field of 15 mT was applied to induce uniaxial anisotropy of the CoFeB layers. We deposited metal layers with a DC power of 30 W and a working pressure of 3 mTorr and a MgO layer with a RF power of 75 W and a working pressure of 10 mTorr. The aforementioned thickness of each layer is a nominal value calculated from the deposition rate of the layer, which was measured using a surface profiler (α-step) and an atomic force microscope. A thin NiFe layer was introduced underneath the IrMn layer to promote the exchange bias of the IrMn/CoFeB bilayer. After the film deposition process, we defined a dumbbell-shaped bottom electrode using photolithography and Ar ion milling techniques. The width of the bottom electrode is 10 μm. Then, pillar-shaped MTJs with a diameter of 5 μm were patterned by etching the top electrode and MgO tunnel barrier and in situ passivating a RF-deposited 70 nm thick SiO_x_ layer to electrically disconnect the bottom and top electrodes. Finally, the top electrode was formed by the deposition of a Cr (5 nm)/Au (100 nm) layer and subsequent lift-off process. Figure 1b presents a scanning electron microscope image of the MTJ device. After the fabrication of the device, MTJs were annealed at different temperatures (*T*_ann_) ranging from 250 °C to 450 °C for 40 min in a vacuum condition. Here, the maximum *T*_ann_ of 450 °C was chosen by considering the thermal budget of the CMOS backend process. To establish exchange coupling at the IrMn/CoFeB interface, the annealing was conducted under a magnetic field of 100 mT, which is sufficient to saturate the magnetization of the CoFeB layer. The tunneling resistance was measured using a four-point geometry method with a constant reading current to apply a bias voltage of 10 mV at room temperature while sweeping the in-plane magnetic fields (*B*_in_). The microstructures of the samples were analyzed through high-resolution scanning transmission electron microscopy (HR-STEM), energy dispersive X-ray spectroscopy (EDS), and X-ray diffraction (XRD).

## 3. Results and Discussion

First, we investigated the annealing effect on the TMR of an MTJ with a Pt (5 nm) capping layer. Figure 1c presents the TMR ratio versus the *B*_in_ curves measured for as-deposited and annealed samples at different *T*_ann_. Here, the TMR ratio is defined as TMR (%) = [$R_{T}$ (*B*_in_) − $\;R_{T}$ (*B*_in_ = 100 mT)]/[$R_{T}$ (*B*_in_ = 100 mT)], where $R_{T}$ is the tunnel resistance of the sample. Note that the magnetization directions of the two CoFeB layers are aligned in a parallel manner when *B*_in_ = 100 mT. The TMR curves show a clear distinction between low and high resistance states for the samples for which *T*_ann_ is below 350 °C, corresponding to the parallel and antiparallel alignment of the top and bottom magnetizations. However, it is severely modified for the sample with *T*_ann_ equal to 450 °C. Figure 1d shows the TMR as a function of *T*_ann_ when it ranges from 200 °C to 450 °C. This finding demonstrates two notable points. First, the TMR ratio of the MTJ with the Pt capping layer can only reach ~95% at *T*_ann_ = 350 °C, much lower than that of typical MgO-based MTJs [12]. Second, the TMR decreases drastically when *T*_ann_ exceeds 350 °C, which may be related to the modified TMR curve.

We also measured the magnetization curves of MTJs annealed at different *T*_ann_ by means of vibrating sample magnetometry. Figure 1e shows two hysteresis curves, one centered at *B*_in_ = 0 and the other centered at *B*_in_ < 0. The former (latter) corresponds to the magnetization of the top magnetically free (bottom exchange-biased) CoFeB layer. Notably, the hysteresis curve of the MTJ with *T*_ann_ = 450 °C is quite different from those of the other samples, a finding consistent with the TMR curve. Figure 1f shows the extracted exchange-bias field (*B*_EB_) of the bottom CoFeB layer and coercivity ($B_{C}^{\mathrm{top}}$) of the top CoFeB layer as a function of *T*_ann_. Here, *B*_EB_ initially increases upon annealing at a *T*_ann_ of 200 °C, after which it gradually decreases upon a further increase in *T*_ann_. We can understand the decrease in *B*_EB_ at a higher *T*_ann_ in terms of the diffusion of Mn from the antiferromagnet IrMn layer [25,26,27]. The behavior of *B*_EB_ with *T*_ann_ is not similar to that of the TMR, and the magnitude of *B*_EB_ still shows a finite value even after annealing at a *T*_ann_ of 450 °C, suggesting that the decrease in *B*_EB_ is not the main cause of the reduction in the TMR ratio at a high *T*_ann_. On the other hand, the $B_{C}^{\mathrm{top}}$ shows different behavior; the values remain unchanged for both the as-deposited and the annealed samples with *T*_ann_ values up to 350 °C. However, it increases drastically upon annealing when *T*_ann_ exceeds 400 °C, with the magnitude becoming ~24 mT, which is more than 10 times greater than those of the samples with lower *T*_ann_s. It was also found that the magnetization of the top CoFeB layer decreases slightly after annealing at a *T*_ann_ of 450 °C. This, together with the large increase in $B_{C}^{\mathrm{top}}$, which causes the antiparallel magnetic alignment to be less pronounced, may be responsible for the reduced TMR ratio.

To understand the *T*_ann_ dependence of the magnetic properties of the top CoFeB layer and the associated TMR ratio, we conducted microstructural analyses using cross-sectional STEM and EDS. Figure 2a,b correspondingly show bright-field STEM images of the as-deposited sample and its annealed counterpart when *T*_ann_ = 450 °C. Both images show clear interfaces of the IrMn/bottom CoFeB/MgO/top CoFeB layers. However, we cannot clearly distinguish the interface between the top CoFeB and Pt layers in either film. To clarify the interface quality, we also investigated high-angle annular dark-field (HAADF) images, in which the contrast is correlated with the atomic number. Figure 2c,d present HAADF-STEM images of the as-deposited and annealed films, respectively. The as-deposited film exhibits a contrast difference, albeit weak, between the top CoFeB and Pt layers. However, the annealed film at a *T*_ann_ of 450 °C shows no contrast difference between the layers. This indicates significant atomic intermixing between the CoFeB/Pt layers.

To assess this intermixing, we construct element mapping images by EDS. Figure 2e,f show images of the as-deposited and annealed films, respectively. Here, yellow, green, and blue represent Pt, Co, and Fe atoms, respectively. It is observed that yellow appears within the top CoFeB layer, even in the as-deposited sample (Figure 2e), indicating that a nonnegligible amount of Pt diffused into the top CoFeB layer during the deposition of the films. The yellow contrast in the top CoFeB layer becomes significantly stronger in Figure 2f, showing that more Pt atoms diffused into the top CoFeB layer by annealing. Note that the contrast of the Co and Fe atoms is not noticeably changed by annealing. We consider Pt diffusion to be the main cause of the reduced TMR. The alloy formation of the diffused Pt with CoFeB can explain the annealing-induced changes in the magnetic properties: an increase in $B_{C}^{\mathrm{top}}$ and a decrease in the magnetization of the top CoFeB layer. In addition to increasing $B_{C}^{\mathrm{top}}$, which makes it difficult to distinguish between parallel and antiparallel magnetization states, Pt diffusion may cause other mechanisms that, in turn, decrease the TMR. One may modify the band structure of CoFeB by adding nonmagnetic Pt, reducing the spin polarization value of the CoFeB layer [46]. Another is that Pt with a face-centered cubic (fcc) structure may prevent CoFeB from being epitaxially matched with the body-centered cubic (bcc) MgO (001) structure, reducing the degree of coherent tunneling [39]. Note that the diffusion of Pt into CoFeB must be suppressed in order to utilize Pt as a spin current source in SOT-based spintronic devices. The interdiffusion within the Pt/CoFeB layers can be mitigated by employing rapid thermal annealing (RTA), which reduces the annealing time [47], and by introducing an insertion layer between the Pt and CoFeB layers. The insertion layer must serve as a diffusion barrier while also enabling the transfer of the spin current generated in the Pt layer to the CoFeB layer without a significant loss [28,31,48].

Next, we examine other capping layers of Ta or W that were introduced between the CoFeB and Pt layers while keeping the remaining layers the same. The layer structure is Ni_81_Fe_19_ (1 nm)/Ir_25_Mn_75_ (15 nm)/Co_32_Fe_48_B_20_ (5 nm)/MgO (2 nm)/Co_32_Fe_48_B_20_ (4 nm)/W or Ta (5 nm)/Pt (5 nm). Hereafter, we refer to the MTJs with a Ta (W) capping layer as Ta-(W-)-capped MTJs. Figure 3a,b show the TMR curves measured at *T*_ann_ of 400 °C and 450 °C for the Ta- and W-capped MTJs, respectively. Note that Ta and W are extensively utilized as spin-current sources, akin to Pt in SOT-based devices [42,43,49,50]. Unlike the MTJ with a Pt capping layer, the TMR curves of these samples retain their shape with well-defined parallel and antiparallel magnetization states, even after annealing at 450 °C. Figure 3c presents the TMR value as a function of *T*_ann_ when it ranges from 200 °C to 450 °C, demonstrating a significant improvement in the TMR ratio and the corresponding *T*_ann_ dependence compared to the Pt-capped MTJs. The maximum TMR ratios become 250% and 190% for the Ta- and W-capped MTJs, respectively. Moreover, *T*_ann_ leading to the maximum TMR increases to 400 °C for both samples. The slight decrease in the TMR ratio at *T*_ann_ = 450 °C may be due to Mn diffusion from the bottom IrMn layer at a high *T*_ann_ [25,26,27]. Figure 3d,e show the magnetization curves of the Ta- and W-capped MTJs films, respectively, with different *T*_ann_s of 400 °C and 450 °C. Unlike the sample with a Pt capping layer (Figure 1d), these samples exhibit two clearly distinguished hysteresis loops for all *T*_ann_s. Figure 3f shows the extracted *B*_EB_ of the bottom CoFeB and $B_{C}^{\mathrm{top}}$ as a function of *T*_ann_. The *B*_EB_ behavior with regard to the *T*_ann_ of the samples with the Ta and W capping layers is very similar to that of the Pt sample; *B*_EB_ initially increases upon annealing at a *T*_ann_ of 200 °C, followed by a gradual decrease with a further increase in *T*_ann_. This result confirms that the change in *B*_EB_ with *T*_ann_ is not the main cause of the *T*_ann_-dependent TMR. On the other hand, the $B_{C}^{\mathrm{top}}$ values do not significantly change for the two samples over the entire *T*_ann_ range up to 450 °C. This is in stark contrast to the Pt sample (Figure 1e). In addition, the magnetizations of the top CoFeB layer of the W- and Ta-capped samples remain unchanged after annealing at a *T*_ann_ of 450 °C. These results demonstrate that the magnetic properties of the top CoFeB with the Ta or W capping layer are robust to the annealing at a *T*_ann_ of up to 450 °C, possibly because of the suppressed atomic diffusion. This is in line with the enhancement of the *T*_ann_ dependence of the TMR.

Additionally, we investigated the microstructures of Ta-capped MTJ films upon annealing using cross-sectional STEM and EDS. Figure 4a,b show bright-field STEM images of as-deposited and annealed (*T*_ann_ = 450 °C) films, respectively. Both images reveal well-defined layers and sharp interfaces, including the top CoFeB/Ta layers. The HAADF-STEM images as shown in Figure 4c,d exhibit a clear contrast difference for all layers, confirming that the top CoFeB/Ta layer structure is maintained after annealing at *T*_ann_ = 450 °C. Figure 4e,f show EDS element mapping images of the as-deposited and annealed films, respectively. Here, red, green, and blue represent Ta, Co, and Fe atoms, respectively. No red color is observed on the top CoFeB layer, indicating that the diffusion of Ta atoms into the top CoFeB layer after annealing at 450 °C was suppressed. This result is in stark contrast to that of the Pt-capped MTJ shown in Figure 2. On the other hand, the green and blue contrasts corresponding to Co and Fe atoms did not change significantly after annealing, similar to the Pt-capped MTJs. The W capping layer shows diffusion behavior similar to that of the Ta layer (Appendix A). Here, the Pt capping layer more easily intermixes with CoFeB than the Ta or W capping layer, possibly because of the greater solubility of Pt in Co or Fe compared to W or Ta in Co or Fe [51,52,53,54,55,56]. Furthermore, X-ray photoemission spectroscopy measurements reveal that the Ta layer acts as an effective boron absorber during the annealing process, thereby facilitating the crystallization of the CoFeB layer (Appendix A). These results indicate that the Ta capping layer effectively serves as an appropriate capping layer that suppresses atomic diffusion during the annealing process, consequently enhancing the *T*_ann_ dependence of the TMR and satisfying the thermal budget requirement of the CMOS backend process.

We also examined microstructural changes upon annealing using the high-resolution *θ*–2*θ* XRD measurements. Note that we used thicker MgO (10 nm) and top CoFeB (10 nm) layers than those in the MTJ devices to enhance the XRD signal. Figure 5a shows the XRD spectra of the Pt samples with different *T*_ann_s of 350 and 450 °C. It was found that the MgO (200) peak appears after annealing with *T*_ann_ = 350 °C and that the corresponding intensity decreases at *T*_ann_ = 450 °C, while the FePt (111) peak appears with 2*θ* = 44.21° [57]. This result provides the microstructural origins of the significant reduction in the TMR ratio at a high *T*_ann_: the degradation of the MgO (001) crystal structure and the formation of a second phase of FePt caused by the diffusion of Pt atoms. On the other hand, Figure 5b,c show the XRD measurement results of the Ta and W samples, where the MgO (200) peaks remain unchanged and no additional peak emerges after annealing at *T*_ann_ = 450 °C. This again confirms that interdiffusion in the Ta and W samples is significantly suppressed even at a *T*_ann_ of 450 °C. These results are consistent with the enhanced temperature dependence of the TMR ratio of the Ta- and W-capped MTJs.

## 4. Conclusions

In this study, we investigated the annealing effect on TMR in IrMn/CoFeB/MgO/CoFeB MTJs with different heavy-metal capping layers. For an MTJ with a Pt capping layer, the TMR reaches a maximum value of ~95% with *T*_ann_ = 350 °C, decreasing drastically when *T*_ann_ is increased further. With microstructural analyses using STEM and EDS, we attributed this low TMR to significant intermixing between the Pt and top CoFeB layers. This suggests that in order for Pt to be used as a SOT material, the property degradation after annealing at elevated temperatures must be overcome. Unlike the Pt capping layer, when introducing Ta and W capping layers with suppressed diffusion into the top CoFeB, the TMR ratio and corresponding temperature dependence significantly improved, showing a maximum TMR of ~250% at *T*_ann_ = 400 °C. Our study highlights the importance of the capping layer, which can significantly affect the MTJ device performance capabilities, suggesting that a proper capping layer will enhance the temperature dependence of the TMR so that applicability to the CMOS backend process can be realized.

## Figures and Tables

**Figure 1 nanomaterials-13-02591-f001:**
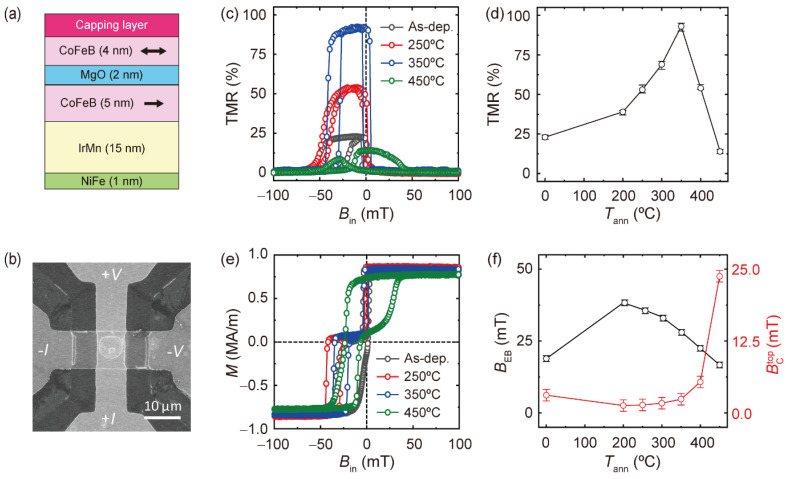
Annealing-temperature (*T*_ann_)-dependent TMR and magnetic properties in MTJs with a Pt capping layer. (**a**) Schematic of an MTJ with an IrMn/CoFeB/MgO/CoFeB structure. (**b**) Scanning electron microscope image of the MTJ device. (**c**) TMR curves of MTJs with different *T*_ann_. (**d**) TMR as a function of *T*_ann_. The error bars are obtained by averaging the measurement results from three MTJ devices. (**e**) Hysteresis loops of the MTJ films with different *T*_ann_. (**f**) Exchange-bias field of the bottom CoFeB layer (*B*_EB_) and coercivity of the top CoFeB layer ($B_{C}^{\mathrm{top}}$) as a function of *T*_ann_. The error bars represent the uncertainties caused by the resolution of the measurement systems.

**Figure 2 nanomaterials-13-02591-f002:**
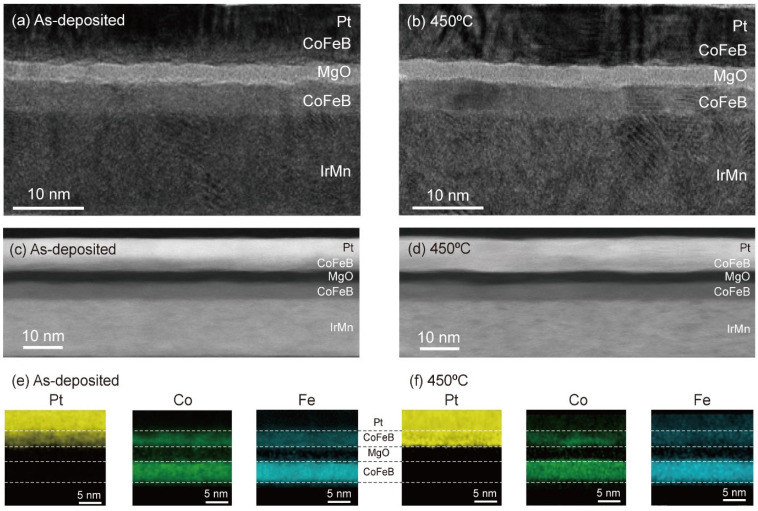
Microstructural analysis using cross-sectional STEM and EDS. (**a**,**b**) Bright-field STEM images of a MTJ with a Pt capping layer; as-deposited (**a**) and annealed films (**b**). (**c**,**d**) HAADF-STEM images of as-deposited (**c**) and annealed (**d**) films. (**e**,**f**) EDS element mapping images of Pt, Co, and Fe atoms; as-deposited (**e**) and annealed films (**f**). *T*_ann_ = 450 °C.

**Figure 3 nanomaterials-13-02591-f003:**
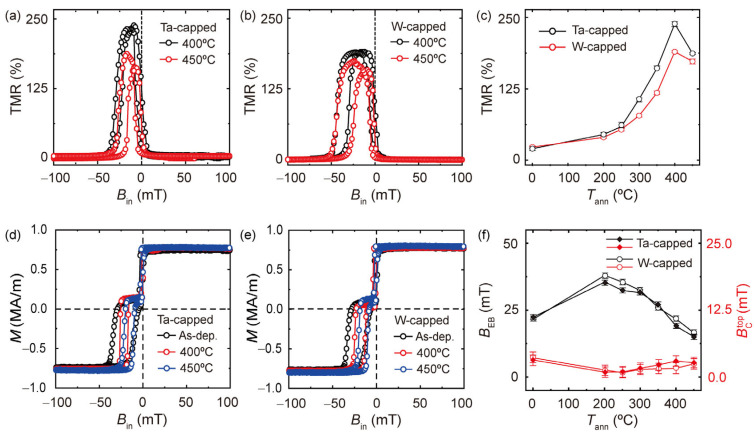
Annealing-temperature (*T*_ann_)-dependent TMR and magnetic properties in MTJs with Ta and W capping layers. The layer structure is Ni_81_Fe_19_ (1 nm)/Ir_25_Mn_75_ (15 nm)/Co_32_Fe_48_B_20_ (5 nm)/MgO (2 nm)/Co_32_Fe_48_B_20_ (4 nm)/W or Ta (4 nm)/Pt (5 nm). (**a**,**b**) TMR curves of MTJs with a Ta capping (**a**) and a W capping (**b**) layer annealed at a *T*_ann_ of 400 °C and 450 °C, respectively. (**c**) TMR as a function of *T*_ann_ in Ta- and W-capped MTJs. The error bars are obtained by averaging the measurement results from three MTJ devices. (**d**,**e**) Hysteresis loops of the Ta-capped (**d**) and W-capped (**e**) MTJ films with different *T*_ann_. (**f**) Exchange-bias field of the bottom CoFeB layer (*B*_EB_) and coercivity of the top CoFeB layer ($B_{C}^{\mathrm{top}}$) as a function of *T*_ann_ for the Ta- and W-capped MTJ films. The error bars represent the uncertainties caused by the resolution of the measurement systems.

**Figure 4 nanomaterials-13-02591-f004:**
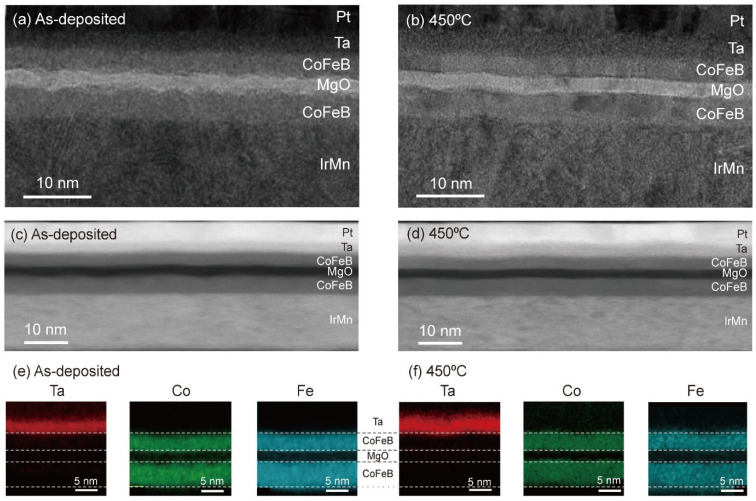
Bright-field STEM images of the Ta-capped MTJ structure of as-deposited (**a**) and annealed (**b**) films. The layer structure is Ni_81_Fe_19_ (1 nm)/Ir_25_Mn_75_ (15 nm)/Co_32_Fe_48_B_20_ (5 nm)/MgO (2 nm)/Co_32_Fe_48_B_20_ (4 nm)/Ta (4 nm)/Pt (5 nm). (**c**,**d**) HAADF-STEM images of the Ta-capped MTJ structure of as-deposited (**c**) and annealed (**d**) films. (**e**,**f**) EDS element mapping images of as-deposited (**e**) and annealed (**f**) films when *T*_ann_ = 450 °C.

**Figure 5 nanomaterials-13-02591-f005:**
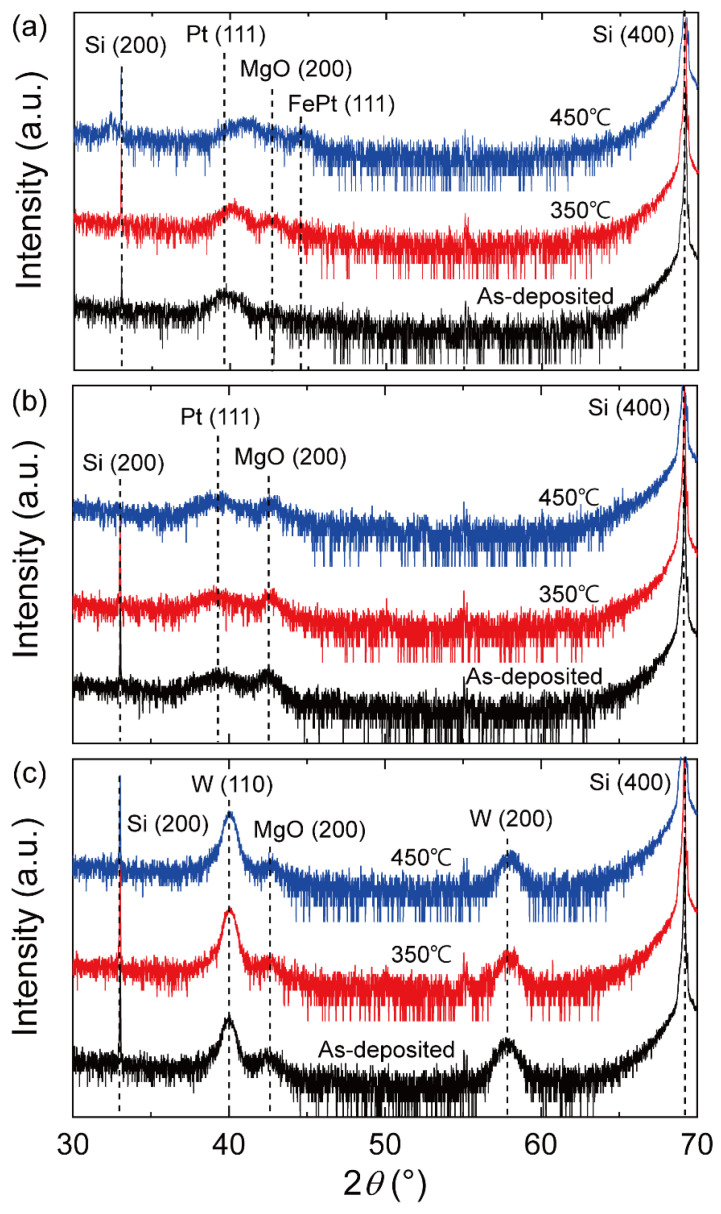
(**a**–**c**) The X-ray diffraction spectra of the samples with different capping layers. The film consists of a Ta (2 nm)/CoFeB (3 nm)/MgO (10 nm)/CoFeB (10 nm) structure with Pt (5 nm) (**a**), Ta (4 nm)/Pt (3 nm) (**b**), and W(4 nm)/Pt (3 nm) (**c**). *T*_ann_ is 350 °C and 450 °C.

## Data Availability

Data are available from the correspondence author.

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
