# Peer review of "The Influence of Capping Layers on Tunneling Magnetoresistance and Microstructure in CoFeB/MgO/CoFeB Magnetic Tunnel Junctions upon Annealing"

_nanomaterials, 2023, doi:10.3390/nano13182591_

Round 1

Reviewer 1 Report

Dear Editor,

The manuscript “The role of capping layers on tunneling magnetoresistance and microstructure in CoFeB/MgO/CoFeB magnetic tunnel junctions upon annealing” by Geunwoo Kim, Soogil Lee, Sanghwa Lee, Byonggwon Song et al, studies experimentally influence of fabrication parameters of the tunneling magnetic junction structure on the junction performance, namely the TMR value. Authors tried few capping layers and see how the annealing temperature affect the TMR effect. The main finding is that Ta capping layer allows to increase the annealing temperature and get it into the CMOS fabrication process range without reducing of TMR effect.

My main concern about this work is it’s novelty. Authors claim in the introduction that “the role of the capping layer in contact with the upper interface of the CoFeB layer during a heat treatment has been studied relatively less thus far [27-30], as the capping layer has mainly been used to prevent the oxidation of FM.” They provide only 4 references here. Actually, there are much more references on the capping layer importance out there. Here I provide just few that are not mentioned in the manuscript (Appl. Phys. Lett. 103, 142412 (2013), Appl. Phys. Lett. 110, 202401 (2017), Journal of Applied Physics 122, 103904 (2017), Journal of Applied Physics 105, 07C915 (2009), Appl. Phys. Lett. 106, 182406 (2015), Journal of Applied Physics 109, 07C711 (2011)). So, the role of the capping layer was actually extensively studied. The second half of the statement is also incorrect. In contrast to authors statement, people understand that capping layer is not just to prevent the oxidation. It was demonstrated that capping layer affects the TMR ratio. There are few factors here: the strain produced by the capping layer, boron diffusion from CoFeB to the capping layer and intermixing. All these factors were discussed before. Also, the annealing temperature dependence of the TMR was studied before in the connection to the capping layer influence. Authors actually do not discuss what people studied before and what people understand on the capping layer importance. Therefore, the novelty of the manuscript and the place of the work among other works is not clear. Authors should rework the introduction and explain what is different in their work comparing to previous ones. So, I suggest not to publish the manuscript in the present form.

Other comments:

1.       It is not clear why authors made an investigation of the W capping layer but does not mention it in the conclusion and abstract? Note that the TMR effect is quite strong in this system as well, especially at high temperature (450C). That would be instructive to provide elements distribution for the W-capped junctions as well.

2.       Ca authors also provide the distribution of boron before and after annealing? It is well known that TMR variation is related to boron diffusion from CoFeB to capping layer.

Minor comments:

1.       The notation Ta – annealing temperature is easy to confuse with notation of tantalum (Ta) in the manuscript. Another notation of the annealing temperature should be chosen like T_an. Or at least “a” should be made as a subscript.

2.       The font in lines 98-100 is different. Should be fixed.

Language is fine. Minor changes are needed.

Author Response

We attached the response letter.

Reviewer 2 Report

The authors report magneto-transport properties and microstructure of magnetic tunnel junctions (MTJs) consisting of a CoFeB/MgO/CoFeB multilayer capped by Pt, Ta or W. It is shown that the Ta-capped MTJs exhibit superior tunneling magnetoresistance (TMR) ratios compared with the Pt-capped MTJs owing to the smaller atomic intermixing at the CoFeB/Ta interface. 

The Ta-capped in-plane magnetized MTJs as reported by the authors has been intensively studied since the discovery of large TMR in CoFeB/MgO/CoFeB MTJs (D.D. Djayaprawira et al., APL 86, 092502 (2005)). The TMR ratio obtained in this work is comparable or even smaller than previous studies (e.g., S. Ikeda at al., APL 93, 082508 (2008)), and more detailed nanostructural/compositional analyses for the Ta/CoFeB interface with varying the annealing temperature can be found in various papers including H. Bouchikhaoui et al., Acta Mater. 116, 298 (2016), and X.D. Xu et al., Acta Mater. 161, 360 (2018). It should be also pointed out that their claim of “when introducing a Ta capping layer with negligible diffusion into the top CoFeB” is quite misleading because the diffusion of Ta cannot be neglected for state-of-the-art perpendicularly magnetized MTJ devices using ultrathin CoFeB films (see, for example, T. Yamamoto et al., Acta Mater. 216, 117097 (2021)). 

Overall, I cannot find any advance in the field of spintronics and/or electronic nanomaterials brought by the present study. 

Author Response

We attached the review report. Please see the attached file.

Reviewer 3 Report

The current work investigates the role of annealing and capping layers (Ta & Pt & w) in enhancing the magnetoresistance response of well-known CoFeB/MgO/CoFeB-based magnetic tunnel junctions (MTJs). The authors observed a change in the TMR percentage values by changing the annealing condition and capping layer. The novelty of the current work is on average and the effect of the capping layer and the annealing condition of magneto-structural properties CoFeB/MgO/CoFeB-based magnetic tunnel junctions (MTJs) was well-investigated by many research groups. The current work needs extensive enhancement before publishing in Nanomaterials.

1-     The introduction needs to be enhanced with more studies investigating the effect of sample-preparing parameters, especially since this system is very well known. In addition, it will be great if they spot line the gap in this research point and the current study's importance to fill it.

2-     The experimental part needs to enhance with more details about the film deposition and how the authors estimate the layer thickness of the film. Also, the authors should add in the revised version information about fabricated Pillar-shaped MTJs by using photolithography. In addition, the author should explain why they chose these annealing conditions and applied specific values of the magnetic field. If they have studied these parameters before? Please, illustrate this point.

3-     The authors should add the M-H loops for samples with capping layers Ta and W like Figures 1d & 1e and their related discussion to have a complete view of the effect of the different capping layers on TMR and M-H.

4-     It will be great if the authors add XRD information and analysis of all phases that can be induced by different annealing temperatures for different capping layers.

5-     The authors should add the uncertainties for all data obtained such as layer thickness, coercivity and EB estimation.

6-     It is better for using different abbreviations of temperature annealing (Tann) to avoid mixing with Ta capping layer.

The English needs to be improved in the whole of the manuscript.

Author Response

(The authors gave the same response as above.)

Reviewer 4 Report

The article is devoted to the actual problem of changing electromagnetic properties in CoFeB/MgO/CoFeB magnetic tunnel junctions upon annealing.

However , when reading , the following questions arise:

1) Which devices are supposed to use the entire Ni81Fe19 (1 nm)/Ir25Mn75 (15 nm)/Co32Fe48B20 (5nm)/MgO (2 nm)/Co32Fe48B20 (4 nm) nanostructure?

2) What is the role of the lowest layers of Ni81Fe19 (1 nm)/Ir25Mn75 (15 nm) in  MTJs ?

3) Why are the results of the influence of the annealing temperature on the magnetic properties of MTJs with Ta and W covering layers not presented, as for Pt in Fig.1d ?

4) Why of the three metals Pt, Ta and W, platinum Pt interacts most strongly with the upper layer of Co32Fe48B20 (4 nm)?

5) What are the microstructure and phase composition of all nanolayers in the initial nanostructures and how do they change during annealing? After all, the amourphic nanolayers Co32Fe48B20 obtained should form nanocrystals upon annealing.

In addition, in the INTRODUCTION, the authors should formulate the purpose of the work, and in CONCLUTION to emphasize the novelty of the results obtained.

And finally, according to the reviewer, in the title of the article should be replaced: "The role ..."

on "The influence of capping layers on tunneling magnetoresistance and microstructure in CoFeB/MgO/CoFeB magnetic tunnel junctions upon annealing".

After making the appropriate additions to the text, the article can be published in the journal "Nanomaterials".

Author Response

(The authors gave the same response as above.)

Round 2

Reviewer 2 Report

In the revised manuscript as well as in the response letter, the authors mentioned the importance of the annealing effect in Pt/CoFeB bilayers especially in terms of the SOT applications. Then, it would be helpful for the readers if the authors can suggest some strategies to improve the electric/magnetic properties of the Pt/CoFeB bilayers after the high temperature annealing, not just comparing the results with the Ta/CoFeB and W/CoFeB cases in which a superior thermal tolerance has been confirmed. 

Although the authors claimed in the revised manuscript that “Note that TMR relies on the CoFeB/MgO interfaces, but PMA depends on the Pt/CoFeB interface. (Page 4, Line 13)”, this is not well supported by their experimental results and references. In the most case, PMA in X/CoFeB/MgO multilayers is dominated by the interfacial PMA at the bcc-CoFeB/MgO achieved by the post-annealing. It is rather unclear whether the use of Pt capping layer enhances the PMA since the use of fcc-Pt inhibits the solid-state epitaxy of bcc-CoFeB at the CoFeB/MgO interface while causing a severe interdiffusion as they have observed in the experiments. 

This paper may be accepted for publication after addressing the above issues. 

Reviewer 3 Report

The author fixed all my comments. The manuscript can publish in the present form. 

Reviewer 4 Report

The authors have significantly supplemented and improved the article. However, there are still the following questions about improving the quality of paper.

1) As follows from Figures 4 and 5 of the article and Figure 3c of the supplement, Ta and W are sublayers of a two-layer coating, and the covering layers are a single-layer Pt coating and two-layer Ta/Pt and W/Pt coatings. And the paper essentially examines the effect of the Ta and W sublayers shielding the upper covering layer Pt from the upper CoFeB layer of the entire MTJ structure on the electromagnetic properties of the latter. This fact should be reflected in the text of the article.

2) Why do we see the reflection (200) from the thinnest layer on the diffractograms in Fig.5

MgO(2 nm), consisting of light elements, and we do not see a single reflection from the thickest layer of IrMn (15 nm)? It cannot be that the thinnest layer of MgO gives the same intense reflection (200) as Pt (111). This reflection at 2θ≈42ο may refer to one of the phases of the CoFe alloy. Check it out!

What phase does the line 2θ≈70ο belong to in all diffractograms and the reflection 2θ≈33ο in Fig.5a 450oC?

After answering these questions, the article can be published in the journal Nanomaterials
